# miRNA-Mediated Epigenetic Regulation of Treatment Response in RA Patients—A Systematic Review

**DOI:** 10.3390/ijms232112989

**Published:** 2022-10-27

**Authors:** Arkaitz Mucientes, Jose Manuel Lisbona, Natalia Mena-Vázquez, Patricia Ruiz-Limón, Sara Manrique-Arija, Antonio Fernández-Nebro

**Affiliations:** 1Instituto de Investigación Biomédica de Málaga y Plataforma en Nanomedicina-IBIMA Plataforma Bionand, 29010 Málaga, Spain; 2UGC de Reumatología, Hospital Regional Universitario de Málaga, 29009 Málaga, Spain; 3Departamento de Medicina y Dermatología, Universidad de Málaga, 29010 Málaga, Spain; 4Unidad de Gestión Clínica de Endocrinología y Nutrición, Hospital Clínico Virgen de la Victoria, 29010 Málaga, Spain; 5CIBER Fisiopatología de la Obesidad y Nutrición (CIBEROBN), Instituto de Salud Carlos III, 28029 Madrid, Spain

**Keywords:** microRNA, rheumatoid arthritis, treatment, biomarker, disease-modifying antirheumatic drug, systemic review

## Abstract

This study aimed to evaluate the role of microRNAs (miRNA) as biomarkers of treatment response in rheumatoid arthritis (RA) patients through a systematic review of the literature. The MEDLINE and Embase databases were searched for studies including RA-diagnosed patients treated with disease-modifying antirheumatic drugs (DMARDs) that identify miRNAs as response predictors. Review inclusion criteria were met by 10 studies. The main outcome of the study was the response to treatment, defined according to EULAR criteria. A total of 839 RA patients and 67 healthy donors were included in the selected studies. RA patients presented seropositivity for the rheumatoid factor of 74.7% and anti-citrullinated C-peptide antibodies of 63.6%. After revision, 15 miRNAs were described as treatment response biomarkers for methotrexate, anti-tumour necrosis factor (TNF), and rituximab. Among treatments, methotrexate presented the highest number of predictor miRNAs: miR-16, miR-22, miR-132, miR-146a and miR-155. The most polyvalent miRNAs were miR-146a, predicting response to methotrexate and anti-TNF, and miR-125b, which predicts response to infliximab and rituximab. Our data support the role of miRNAs as biomarkers of treatment response in RA and point to DMARDs modifying the miRNAs expression. Nevertheless, further studies are needed since a meta-analysis that allows definitive conclusions is not possible due to the lack of studies in this field.

## 1. Introduction

Rheumatoid arthritis (RA) is an immune-mediated inflammatory disease characterised by chronic synovial inflammation, progressive and irreversible joint destruction and functional disability. It has been estimated that RA affects 0.5–1% of the adult population worldwide [1]. Nevertheless, the detailed mechanisms underlying the pathogenesis of RA, disease activity and its severity remain not fully elucidated [2]. Consequently, there is no reliable biomarker for RA diagnosis. A holistic understanding of AR including both genetic and epigenetic (DNA methylation, microRNA and histone modifications) perspectives is needed for early diagnosis and personalised treatment [3,4,5]. Currently, there is no cure for RA. Disease-modifying antirheumatic drugs (DMARDs) are the conventional therapeutic approach to treat RA. In recent years, the therapeutic options were expanded by developing tumour necrosis factor (TNF) inhibitors and other biological agents that have improved both the management and prognosis of RA [6]. Despite these benefits, clinical practice has demonstrated that these biological agents are not effective for all RA patients: 20% to 40% of RA patients discontinue the treatment due to several reasons, such as lack of efficacy in treatment, adverse events or inability to afford its high cost [7,8]. Therefore, identifying reliable markers of response to treatment will benefit the patient’s quality of life and optimise healthcare resources.

Small RNAs (sRNA) are RNA molecules, usually non-coding, and 18–30 nucleotides in length. MicroRNAs (miRNA), the most studied sRNAs, are endogenous, non-coding, single-stranded, highly conserved, and 20–22-nucleotide-long RNAs [9]. To date, miRNA genes are estimated to constitute 1–2% of the complete genome, and more than 2000 miRNAs have been identified [10]. After being transcribed, mature miRNAs interact with their messenger RNA (mRNA) targets by hybridising into complementary sequences in the mRNA 3′-UTR regions [1]. This binding results in translational repression or mRNA degradation, regulating the gene expression [11]. Around a third of the total protein-coding genes are controlled by miRNAs, whereas 60% of the genes present miRNA-binding domains [12]. Thus, miRNAs act as an epigenetic control agent determining gene expression, acquiring a pivotal role in biological processes. miRNAs present good stability, and they can be detected in tissue samples and different biological fluids: blood, serum, saliva, plasma, and urine [9,13]. Furthermore, miRNAs can be determined by relative economic, simple and reproducible assays.

The described characteristics give miRNAs a potential role as biomarkers of diseases in which they have been described as being involved. Thus, miRNAs are used as biomarkers of cancers, cardiovascular and autoimmunity diseases, including RA [14,15,16]. In RA patients, it has been communicated altered levels of miRNA in blood, plasma, synovial fluid and cells lining the joint capsule [17,18,19]. This makes sense, as miRNAs are related to the proliferation and differentiation of inflammatory cytokines, synovial cells and osteoclast [10]. Moreover, miRNAs contribute to inflammation formation and immune response, which are altered in autoimmune diseases [20,21]. Described altered miRNA in RA may be potential biomarkers of the disease state, gravity, prognosis or treatment response: Bhanji et al. described the role of miRNA in RA progression in 2007 [22], and Filkova et al. proposed that both disease severity and duration, or the effect of treatment, could modulate the levels of circulating miRNAs in established RA patients [23]. 

All these data indicate a major role of the miRNAs in the RA epigenome. As stated above, markers of treatment response are needed. Therefore, it is plausible to consider miRNA levels as biomarkers of treatment response in the RA context. In recent years, several studies have tested miRNAs as biomarkers of both conventional synthetic DMARDs (csDMARDs) and biological DMARDs (bDMARDs). All the studies referenced present differences in methodology and results [1,13,24,25,26,27,28,29,30]. Hence, we aim to systematically review and resume the recent literature that analyses the role of miRNAs as biomarkers of treatment response in RA patients and provide an overview of the current literature. To our knowledge, this is the first work addressing this objective.

## 2. Methods

### 2.1. Search Strategy and Studies Selection

A systematic search was conducted for papers studying the expression of miRNAs in RA in relation to the therapeutic response in relation with the therapeutic response at both Medline and Embase databases using the following MeSH terms, Entry Terms and text-free: “Rheumatoid arthritis” and “miRNA” or “microRNA” or “microRNAs” and “variant” or “variants” or “mutation” or “polymorphism” or “polymorphisms” (Appendix A). Moreover, a secondary manual search of the related articles was also performed. It was restricted to the English language and human studies. The review protocol followed the declaration of the Preferred Reporting Items for Systematic Reviews and Meta-Analyses (PRISMA) statement (Figure 1). The research question was “Can miRNAs be useful as biomarkers of therapeutic response in RA?” and it was structured following the PICOS methodology (population, intervention, comparison, outcomes, and study design). The search was performed by two researchers (A.M. and J.M.L.) who independently reviewed article titles and abstracts. Disagreement between reviewers on the inclusion/exclusion of studies was solved by consensus or with the assistance of a third reviewer (N.M-V.). 

### 2.2. Inclusion and Exclusion Criteria

These were the inclusion criteria for this review: (1) English language; (2) clinical trials, transversal studies, and case-control studies including adult RA patients according to ACR/EULAR classification criteria [17] treated with csDMARDs or bDMARDs; and (3) studies identifying miRNAs. Exclusion criteria were (1) editorials, narrative reviews, congress abstracts, case reports or case series with less than 30 cases; (2) inadequate description of methodology; (3) lack of data to evaluate the response to treatment; and (4) duplicated publications.

Regarding inclusion criterion 3, only studies that determine the specific association between miRNAs and treatment response were included. Thus, studies that communicated only the association between specific miRNAs with other RA characteristics were excluded. 

### 2.3. Outcome Measures

The main outcome was a good therapeutic response measured by the disease activity score using 28 joint counts (DAS28) [31]. Response to treatment was defined according to EULAR criteria, based on the changes in the DAS28 score: an improvement in DAS28 over ≥1.2 and a DAS28 value ≤ 3.2 after treatment was considered a good response; a DAS28 value between 3.2 and 5.1 and a reduction between 0.6 and 1.2 was considered a moderate response; and patients with a DAS28 score > 5.1 or a reduction in DAS28 under 0.6 were considered non-responders [32].

### 2.4. Data Extraction and Measures of Study Quality

The whole text was read in the case of articles whose titles or abstracts met the inclusion criteria. Not fulfilling an eligibility criterion led the study to being excluded. In addition to the main and secondary outcome measures, we extracted information on the authors, year of publication, sex, age, autoantibodies, treatment groups, treatment response, miRNA associated and sample types of the patients. In order to evaluate the quality of the studies, the level of the evidence was assessed using the Scottish Intercollegiate Guidelines Network (SIGN) grading system [33].

## 3. Results and Discussion

### 3.1. Search

A systemic search was carried out, and 698 articles published between 1998 and 2022 were identified. Of them, 44 were duplicated and therefore eliminated from the study. After checking both the title and abstract, 628 articles were excluded. Finally, 13 articles were excluded after being fully read due to different causes (Appendix A). Thus, 10 articles fulfilled the established inclusion criteria that constitute the focus of this systematic review (Figure 1).

### 3.2. Characteristics of the Included Studies

Eight observational studies and two clinical trials were included. The SIGN grading system results confirmed that all included studies present a low or very low bias risk (Appendix A). Table 1 resumes the general characteristics of the included studies. Among the studies, 1/10 analysed the whole microRNome [24]; 5/10 assayed arrays of a different number of miRNAs—377 [26], 91 [28], 84 [27], 758 [13], and undetermined [1]; finally, 4/10 opted for a fine-mapping strategy by analysing a low number of specific miRNAs [25,29,30,34]. Regarding the type of sample, in all studies, peripheral blood was collected and, if needed, processed to obtain plasma, serum or peripheral blood monocellular cells before miRNA isolation: 2/10 studies used plasma samples, 4/10 studies used serum samples, 2/10 studies used whole blood samples, 1/10 both whole blood and plasma samples, and 1/10 used peripheral blood monocellular cells. Therefore, all studies analysed levels of circulating miRNAs.

The main characteristics of the patients are shown in Table 1. A total of 839 RA patients were included in the 10 studies after taking into account that 2/10 studies presented the same 108 RA patients, belonging to the OPERA cohort [35], but analysed different miRNAs [26,27]. Among the studies, 2/11 presented discovery and validation cohorts [13,28]; these cohorts were composed of 90 and 125 RA patients, respectively. Only in 3/10 studies, a healthy donor group was included [30,34], adding a total of 67 healthy subjects. Moreover, 2/10 studies included patients with different conditions: 13 ankylosing spondylitis patients, and 13 B lymphoma patients in addition to RA patients [30,34]. Overall, the female sex was predominant (74.5%), and the mean ± SD age was 53.5 ± 5.3 years. All studies give information about RA patients’ autoantibodies, and only Ciechomska et al. do not indicate if RA patients had previously had an inadequate response to at least one DMARD and/or were naïve to treatment (Appendix A). Most RA patients presented seropositivity for rheumatoid factor (RF) of 74.7% and anti-citrullinated C-peptide antibodies (ACPA) of 63.6%. Moreover, patients included in the study of Luque-Tevar et al. [24] showed elevated titres for RF (112.9 ± 205.8) and ACPA (343.3 ± 762.6). Finally, information about glucocorticoid administration was lacking in 3/10 studies (Appendix A). Only Cuppen et al. verified whether glucocorticoids have an effect on treatment response [13]. Their results showed a significant difference between responders and non-responders regarding the use of glucocorticoids in adalimumab (ADA) treatment (responders: 13%, non-responders: 50%, *p* = 0.01), but not in etanercept (ETN) treatment (responders: 20%, non-responders: 33%, *p* = 0.25).

### 3.3. miRNAs as Indicators for Treatment Response

Table 2 shows the microRNAs described as associated with the treatment response in the studies included in the present systemic review. These miRNAs were tested as biomarkers for both csDMARDs and bDMARDs.

An important characteristic in RA biologic treatment is whether the patient has shown an inadequate response to other(s) biologic treatment(s) or the patient is naïve to biologic drugs. Half of the included studies used a cohort composed of RA patients naïve to bDMARDS. Hence, Singh et al., after studying the expression in 94 RA patients, found that miR-132, miR-146a and miR-155 are lowly expressed in MTX-responder RA patients [25]. Luque-Tevar et al., with a cohort composed of 104 RA patients and 29 healthy donors, observed that elevated levels of miR-106a were associated with a good response to several anti-TNF treatments [24]. In a previous study with a similar design, Castro-Villegas et al. informed that serum levels of miR-23 and miR-223 can predict response to anti-TNF treatment [28]. In their randomised clinical trial including treatment-naïve RA patients, Krintel et al. associated the low expression of miR-22 combined with miR-886 high expression with a good response to adalimumab (ADA), combined with methotrexate (MTX) [26], whereas Sode et al. concluded that higher expression of miR-27a prior treatment is associated with remission at 12 months in patients treated with the combination of ADA-MTX in their double-blinded placebo-controlled trial using the same cohort [27]. On the other hand, 4/10 of the studies included RA patients with inadequate response to at least one bDMARD, mostly anti-TNF. The initial results obtained by Cuppen et al. indicated that the expression of miR-99 and miR-143 predicts the ADA response, while the expression of miR-23a and miR-197 indicates the etanercept response, but no association kept the signification level in the validation cohort [13], whereas Liu et al. compared miRNAs levels between etanercept-responder and non-responder RA patients and validated two miRNAs as indicators of treatment response: miR-146a was overexpressed and let-7a was down-expressed in responders compared to non-responders [1]. Cheng et al. communicated that both miR-125a and miR-125b were elevated in RA patients compared to healthy controls, and the basal level of miR-125a predicted response after 24-week infliximab (IFX) treatment [29]. bDMARDs other than anti-TNF were also studied; in this sense, the results obtained by Duroux-Richard et al. indicated that elevated serum levels of miR-125b at disease flare are associated with good clinical response after 3 months of rituximab (RTX) treatment [34]. Furthermore, this predictive role was not limited to RA patients; it was also determined in B-cell lymphoma patients.

Finally, Ciechomska et al. did not inform about the prior use of bDMARDs, and after studying a small cohort of autoimmune patients, they established that the expression of miR-5196 positively correlates with the response to ADA and ETN combined with MTX in both RA and ankylosing spondylitis patients [30]. 

Disease-modifying antirheumatic drugs (DMARDs) are the conventional treatment for RA. Nevertheless, between 30 and 40% of the patients do not respond to the treatment, which involves maintained inflammatory activity, potential adverse effects, and continuous changes in treatment [36]. Hence, identifying reliable biomarkers for treatment response will have a positive impact on patient life quality and the optimisation of sanitary resources [37]. In this context, miRNAs are emerging as both potential targets for new therapeutic strategies and biomarkers of RA since the association of specific miRNAs with RA has been communicated in the last years [18,19,38]. Based on the dysregulated expression of miRNAs RA in patients, several studies analysing their potential role as biomarkers of treatment response have been published. Considering this, we aim to carry out a synthesis of the literature analysing the role of miRNAs as treatment response biomarkers in the RA context. Thus, the included literature analyses the miRNAs associated with response to csDMARDs [25], csDMARDs/bDMARDs combination [27,28,30] anti-TNFα [1,13,24,26,29], and rituximab [34]. It was not possible to include biomarkers for Jakus kinase inhibitor (JAKi), a promising new drug for RA treatment [39]. To date, no study identifying specific miRNAs as predictors of JAKi response has been published.

The most common DMARDs to treat RA is MTX. Therefore, identifying a reliable biomarker of MTX response is a high priority. In this sense, Singh et al. found that MTX-responder RA patients presented lower baseline levels of miR-132, miR-146a and miR-155, defined as potential biomarkers of responsiveness to MTX treatment [25]. Sode et al., in a placebo-controlled clinical trial, defined miR-27a as a potential biomarker of the response to ADA/MTX treatment, and both miR-16 and miR-22 as biomarkers of the response to MTX treatment [27]. Castro-Villegas et al. concluded that miR-23 and miR-223 are potential biomarkers of anti-TNFα/DMARD combination therapy [28]. Furthermore, in responder patients, the expression of miR-23 and miR-223 correlates with CRP and DAS28, respectively. Finally, Ciemoska et al. [30] observed that levels of miR-5196 decreased after both treatments and correlated with DAS28. These miRNAs are related to the immune and inflammatory response. Several works have communicated their important role in processes such as the regulation of the NF-κB signalling pathway, activation of Th17 cells, mediation of intercellular crosstalk between immune cells, and regulation of IFN-γ, IL-1, IL-4, IL-5, IL-6, Il-8, and IL-12 signalling [38,40,41,42,43,44]. As expected, MTX was the most common DMARD used in the cohorts of all studies. Considering this together with the existing miRNAs dysregulation in RA patients, it is plausible that MTX, alone or combined with anti-TNFα drugs, could restore this dysregulation somehow. Further and specific studies are needed to confirm this hypothesis and describe the underlying mechanisms.

Luque-Tevar et al. evaluated various anti-TNF treatments and concluded that high levels of miR-106a were associated with good response [24]. Likewise, associations have been established with each anti-TNF individually. With ADA, a good therapeutic response has been shown with the overexpression of miR-886 and low expression of miR-22 by Krintel et al. [26], whereas in the study by Cuppen et al. [13], good response was associated with high levels of miR-99a and low levels of miR-143. Regarding ETN, in the study by Liu et al. [1], a good response to treatment was associated with miR-146a, while let-7a had a poor response. The study by Cuppen et al. [13] showed that high levels of miR-23a and miR-197 were associated with a good response. Finally, Cheng et al. [29] studied the response to IFX and determined that miR-125b predicts treatment response at 24 weeks. There is a similar occurrence with RTX; high expression of miR-125b was also observed in responders [32]. Many of the miRNAs evaluated in these studies were found to be increased in response to the treatments used, and it is described that their expression affects the course of RA pathology since among their roles are the release and production of inflammatory factors (such as IL-1β, IL-6, and TNF-α) or the regulation of signalling pathways during inflammation, B cell differentiation, and apoptosis [45,46,47,48]. After treatment, most of the miRNAs evaluated in the analysed works showed a significant alteration in their expression, both in an increasing and decreasing manner, mainly in responder individuals. Since miRNAs are key regulators of gene expression and transcription, these changes are likely to restore the protein dysregulation that RA patients present prior to treatment [34,48]. In this way, previous studies indicated a decreased expression in RA patients in miR-16-3p, miR-132, miR-146a and miR-155 [25]; another study found that that decreased miRNA-146a and 155-3p constituted an abnormal T-regulatory phenotype in these patients [49]. Therefore, this points to an effect of treatment on miRNA expression patterns, but the mechanisms underlying the possible interaction between treatment and miRNAs expression are complex and remain unknown.

In the study by Chen et al. [29], a positive correlation was shown between the basal levels of miRNA-125 with C-reactive protein and of miRNA-125 b with DAS28-ESR in RA patients. In this sense, C-reactive protein and other parameters of disease activity are widely used in the clinical follow-up of patients with inflammatory arthritis and other inflammatory diseases. However, they are nonspecific, as their levels are increased in other inflammatory conditions or remain at normal levels in patients with active disease. Thus, other studies have shown a better correlation and specificity of miRNAs with disease activity than with C-reactive protein [50,51].

Some limitations in this review are related to miRNA research. Nomenclatures used by authors are not consistent since the terms 3p and 5p refer to mature miRNAs, and several studies do not specify whether the results refer to mature miRNAs [52]. Moreover, there is a variety of samples used in the studies: whole blood, serum and plasma. This variety is important, as it is known that there are significant differences in miRNA profiles depending on the tissue [53,54]. Furthermore, it has been established that miRNA expression depends on the stage of RA development [10]. Finally, the number of works aiming to identify miRNAs as biomarkers of treatment response in RA patients found in the literature is lacking, the cohorts included in these studies are small in size, and most of them are not high-quality clinical trials. Furthermore, the treatment guideline significantly differs between studies (e.g., different treatment duration in studies analysing the response to the same treatment). As a consequence of these limitations, it was not possible to carry out a meta-analysis as desirable.

## 4. Conclusions

Overall, our data support the potential role of miRNA as biomarkers of response to different treatments in RA. Moreover, the results point to DMARDs modifying the miRNAs expression, which plays a pivotal role in the modulation of the inflammatory cascade. Three miRNAs stand out due to their polyvalence: miR-146a, which predicts response to MTX and ENT; miR-125b, which predicts response to IFX and RTX; and miR-22, which predicts response to MTX and ADA. Regarding treatments, MTX is the most studied, presenting 5 miRNAs (miR-16, miR-22, miR-132, miR-146a and miR-155) described as response biomarkers. To date, a meta-analysis is not possible to be carry out due mostly to the lacking number of studies. More studies are needed to confirm our results and establish validated predictive models of response to treatment in RA patients.

## Figures and Tables

**Figure 1 ijms-23-12989-f001:**
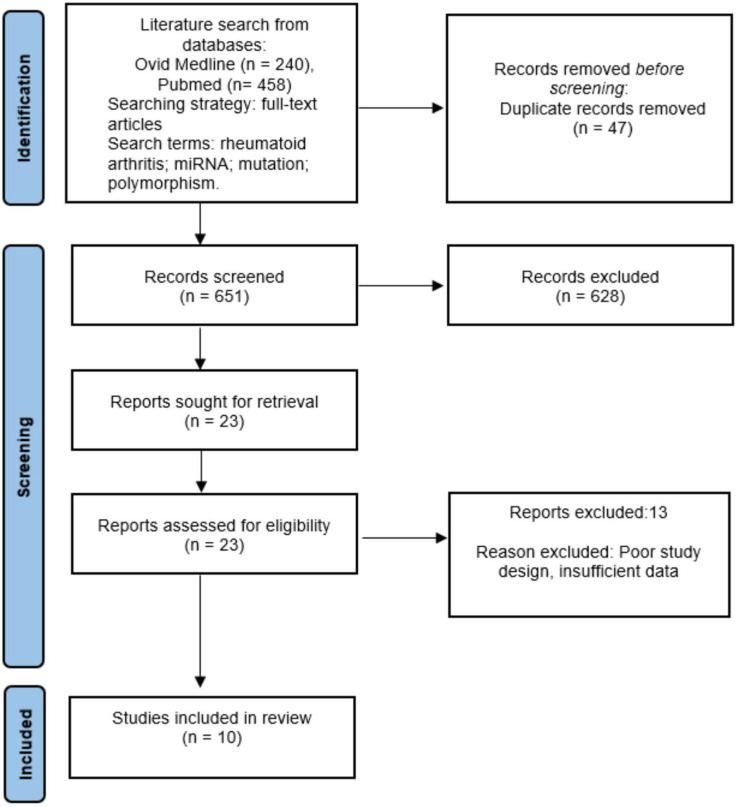
Flow diagram of study selection according to PRISMA statement.

**Table 1 ijms-23-12989-t001:** General characteristics of studies.

Author	Year	Country	microRNAs	Patients	Healthy Donors	Sex (% Women)	Age	Autoantibodies Information	Sample
Luque-Tevar et al. [24]	2012	Spain	Whole miRNome	104 RA patients	29 healthy donors	Patients: 65.5% Controls: 81.0%	Patients (mean ± SD): 47 ± 17.0 Controls (mean ± SD): 51.2 ± 10.5	ACPA, IU/mL (mean ± SD): 343.3 ± 762.6 RF, IU/mL (mean ± SD): 112.9 ± 205.8	Serum from whole blood
Krintel et al. [26]	2015	Denmark	377 miRNAs	180 RA patients (OPERA cohort)	None	Treatment group: 63% Placebo group: 69%	Treatment group: 56.2 (25.8–77.6) Placebo group: 54.2 (28.3–76.7)	Treatment group: ACPA positive: 60% RF positive: 70% Placebo group: ACPA positive: 70% RF positive: 74%	Whole blood
Sode et al. [27]	2017	Denmark	91 miRNAs	180 RA patients (OPERA cohort)	None	Treatment group: 63% Placebo group: 69%	Treatment group: 56.2 (25.8–77.6) Placebo group: 54.2 (28.3–76.7)	Treatment group: ACPA positive: 60% RF positive: 70% Placebo group: ACPA positive: 70% RF positive: 74%	Plasma from whole blood
Singh et al. [25]	2018	India	miR-132 miR-146a miR-155 let-7a	94 RA patients	None	86.2%	Patients (mean ± SD): 40 ± 17	RF positive: 85%	Whole blood
Ciechomska et al. [30]	2018	Poland	miRNA-5196	10 RA patients 13 AS patients	15 healthy controls	RA patients: 60% AS patients: 76.9% Controls: no data	RA patients: 59 (27–74) AS patients: 50 (32–59) Controls: no data	ACPA positive: 30% RF positive: 90%	Serum from whole blood
Castro-Villegas et al. [28]	2015	Spain	84 miRNAs.	Discovery cohort: 10 RA patients. Validation cohort: 85 RA patients.	None	Exploratory cohort: 90%. Validation cohort: 87.1%	Exploratory cohort: 54.6 (38–74) Validation cohort: 53.6 (24–72)	ACPA positive: 66.3% RF positive: 70.5%	Serum from whole blood
Duroux-Richard et al. [34]	2014	France	miR-125b	48 RA patients (32 treated with RTX)	13 healthy donors	84.7%	Patients (mean ± SD): 58.8 ± 7 Controls:	ACPA positive: 82.5%	Whole blood and serum
Cheng et al. [29]	2020	China	miR-125a miR-125b	96 active RA patients	None	80.2%	Patients (mean ± SD): 58.6 ± 10.0	ACPA positive: 62.5% RF positive: 71.9%	Plasma from whole blood
Cuppen et al. [13]	2016	Netherlands	758 miRNAs	RA patients were selected from the BiOCURA cohort. Discovery cohort: 80 RA patients. Validation cohort: 40 RA patients.	None	Discovery cohort: 76.3% Validation cohort: 67.5%	Discovery cohort (mean ± SD): 55 ± 11.0 Validation cohort (mean ± SD): 56 ± 10.0	Discovery cohort: ACPA positive: 71.3% RF positive: 73.8% Validation cohort: ACPA positive: 60% RF positive: 55%	Serum from whole blood
Liu et al. [1]	2019	China	microRNA array	92 active RA patients	None	80%	Patients (mean ± SD): 55.6 ± 8.8	ACPA positive: 77% RF positive: 82%	Peripheral blood mononuclear cells

Abbreviations. miRNAs: MicroRNAs; RA: rheumatoid arthritis. RF: rheumatoid factor; ACPA: anti-citrullinated C-peptide antibodies; AS, ankylosing spondylitis; SD: standard deviation.

**Table 2 ijms-23-12989-t002:** Baseline microRNAs associated with treatment response in RA.

Study	Treatment Groups	Treatment Response	Associated miRNA
Luque-Tevar et al. [24]	(1) IFX (2) ETN (3) ADA (4) GOL (5) CZP	At 3 months: Good: 35.4% Moderate: 31.7% No response: 32.9% At 6 months: Good: 49.4% Moderate: 20.2% No response: 30.4%	High levels of miR-106a were associated with good response.
Krintel et al. [26]	(1) ADA (2) Placebo	At 12 months: Good response (ADA): 72% Good response (Placebo): 63%	The combination of high expression of miR-886 with low expression of miR-22 was associated with a good response.
Sode et al. [27]	(1) MTX-ADA (2) MTX-Saline	At 3 months: Responders (MTX-ADA): 42.7% Responders (MTX-Saline): 24.2% At 12 months: Responders (MTX-ADA): 44.9% Responders (MTX-Saline): 28.6%	High levels of miR-27a were associated with a good response to MTX/ADA. High levels of miR-16 and miR-22 were associated with a good response to MTX
Singh et al. [25]	(1) MTX	At 4 months: Responders: 77.7%	Low levels of miR-132, miR-146a and miR-155 were associated with treatment response.
Ciechomska et al. [30]	(1) MTX-ETN (2) MTX-ADA	At 6 months: Responders: 100%	Expression of miR-5196 correlates with the RA state. (Low-size sample)
Castro-Villegas et al. [28]	(1) IFX (2) ETN (3) ADA	At 6 months: Responders: 89.5%	Expression of miR-23 and miR-223 as biomarkers and predictors of anti-TNFα/DMARDs combination therapy
Duroux-Richard et al. [34]	(1) RTX	At 3 months: Responders (Good/Moderate): 50%.	High expression of miR-125b was associated with good response in active RA patients.
Cheng et al. [29]	(1) IFX	At 24 weeks: Responders: 71.7%	miR-125b predicts treatment response at 24 weeks.
Cuppen et al. [13]	(1) ADA (2) ETN	At 12 months: Responders (ADA): 50% Responders (ETN): 50%	High levels of miR-99a and low levels of miR-143 were associated with ADA response. High levels of miR-23a and miR-197 were associated with ETN response.
Liu et al. [1]	(1) ETN	At 24 weeks: Responders: 65.2%	miR-146a predictive factor for good response. let-7a predictive factor for a worse response.

Abbreviations. RA: rheumatoid arthritis; IFX: infliximab, ETN: etanercept, ADA: adalimumab, GO: golimumab, CZP: certolizumab, MTX: methotrexate, RTX: rituximab.

## Data Availability

Not applicable.

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
