# Peer review of "miRNA-Mediated Epigenetic Regulation of Treatment Response in RA Patients—A Systematic Review"

_ijms, 2022, doi:10.3390/ijms232112989_

Round 1
Reviewer 1 Report
This is a well crafted review that will help the readers. The steps used were clearly described and the rationale explained very well.
Overall, the grammar is okay but can be improved.
Author Response
Comments for the reviewers
We would like to thank the editor for considering our work for publication in “International Journal of Molecular Sciences” and the reviewers for their comments, which have helped to improve the quality of our manuscript.
Below, we provide a point-by-point reply to the comments.
Reviewer #1:
This is a well crafted review that will help the readers. The steps used were clearly described and the rationale explained very well.
Overall, the grammar is okay but can be improved.
Reply: We thank the reviewer for his comment and we have sent the manuscript for grammar checking.
Reviewer #2:
- This is a good paper between micro RNA and DMARDs response in RA. However these micro RNA is more useful than CRP, or disease activity (ex, DAS, CDAI, SDAI,,,) I need some comments about that.
Reply: We appreciate your comment. According to the comments of other reviewers, C-reactive protein and other parameters of disease activity are widely used in the clinical follow-up of patients with inflammatory arthritis and other inflammatory diseases. However, they are nonspecific, as their levels are increased in other inflammatory conditions or remain at normal levels in patients with active disease. Thus, other studies have shown a better correlation of miRNA with disease activity than with C-reactive protein (50,51). For example, in the study by Chen P et al. showed that serum miR-146b-5p correlated significantly with disease activity and was more specific than C-reactive protein (CRP) and other measures of activity (50).
- Chen P, Li Y, Li L, Yu Q, Chao K, Zhou G, et al. Circulating microRNA146b-5p is superior to C-reactive protein as a novel biomarker for monitoring inflammatory bowel disease. Aliment Pharmacol Ther. 2019 Mar;49(6):733–43.
- Schönauen K, Le N, von Arnim U, Schulz C, Malfertheiner P, Link A. Circulating and Fecal microRNAs as Biomarkers for Inflammatory Bowel Diseases. Inflamm Bowel Dis. 2018 Jun;24(7):1547–57.
-Page 9; lines 289-295: “In the study by Chen et al. (29) a positive correlation was shown between the basal levels of miRNA-125 with C-reactive protein and of miRNA-125 b with DAS28-ESR in RA patients. In this sense, C-reactive protein and other parameters of disease activity are widely used in the clinical follow-up of patients with inflammatory arthritis and other inflammatory diseases. However, they are nonspecific, as their levels are increased in other inflammatory conditions or remain at normal levels in patients with active disease. Thus, other studies have shown a better correlation and specificity of miRNAs with disease activity than with C-reactive protein (50,51).”
Reviewer #3:
I have no further comments. The search for biomarkers to predict response to a treatment is of paramount importance in the rheumatology community. This literature review can make a great contribution to further studies on the subject.
Reply: We thank the reviewer for his comment.
Thank you in advance for your time and consideration.
Sincerely yours,
*Correspondence: Natalia Mena Vázquez MD, PhD.
Affiliation: UGC de Reumatología, Instituto de Investigación Biomédica de Málaga (IBIMA)-Plataforma Bionand, Hospital Regional Universitario de Málaga, Málaga, Spain. Plaza del Hospital Civil s/n., 29009 Malaga, Spain. E-mail: nataliamenavazquez@gmail.com
Telephone number/ Fax number: +34 951 290360

Reviewer 2 Report
This is a good paper between micro RNA and DMARDs response in RA. However these micro RNA is more useful than CRP, or disease activity (ex, DAS, CDAI, SDAI,,,) I need some comments about that.
Author Response

(The authors gave the same response as above.)

Reviewer 3 Report
I have no further comments. The search for biomarkers to predict response to a treatment is of paramount importance in the rheumatology community. This literature review can make a great contribution to further studies on the subject.
TRANSLATE with x EnglishArabic | Hebrew | Polish |
Bulgarian | Hindi | Portuguese |
Catalan | Hmong Daw | Romanian |
Chinese Simplified | Hungarian | Russian |
Chinese Traditional | Indonesian | Slovak |
Czech | Italian | Slovenian |
Danish | Japanese | Spanish |
Dutch | Klingon | Swedish |
English | Korean | Thai |
Estonian | Latvian | Turkish |
Finnish | Lithuanian | Ukrainian |
French | Malay | Urdu |
German | Maltese | Vietnamese |
Greek | Norwegian | Welsh |
Haitian Creole | Persian |
Author Response

(The authors gave the same response as above.)

Round 2
Reviewer 2 Report
This paper is accepted by me.